# The Framing of the National Men's Basketball Team Defeats in the Eurobasket Championships (2007–2017) by the Greek Press

Panagiotis Spiliopoulos [1] , Nikolaos Tsigilis [1] , Maria Matsiola [2,*] and Ioanna Tsapari [1]

1. School of Journalism and Mass Communications, Faculty of Economics and Political Sciences, Aristotle University of Thessaloniki, 546 25 Thessaloniki, Greece; panspil@jour.auth.gr (P.S.); ntsigilis@jour.auth.gr (N.T.); tsapariioanna@hotmail.com (I.T.)
2. Department of Communication and Digital Media, University of Western Macedonia, Area Fourka, 521 00 Kastoria, Greece
* Correspondence: mmatsiola@uowm.gr

**Abstract:** As the study of the negatively expressed news on sports constitutes a scientific area that has not received proper attention by researchers yet, the purpose of this study was to investigate the framing of the Greek national men's basketball team defeats by the Greek press. Articles (n = 178) concerning the Eurobasket championships from 2007 to 2017, published in three political newspapers and one sports newspaper, were analyzed through content analysis. Specifically, the research reflected upon (a) the existence of the primary framework of "attribution of responsibility", (b) the differences in framing among the newspapers, and (c) where/to whom the Media focused on regarding the defeats. The "attribution of responsibility" framing was found in all newspapers under study, both in the content and in the titles of the articles, primarily the day after the games. Regarding the defeats, the media were centralized around 12 factors and 8 subfactors as components of responsibilities, while a significant number of other frames were also identified as well as the four stages of the framing function. In conclusion, the finding that framing is significantly met in sports reporting contradicts the credited characterization of "cheerleaders" to sports journalists.

**Keywords:** frames; framing; attribution of responsibility; sports journalism; basketball; Eurobasket championships; Greek national team

## 1. Introduction

For many decades, media have been the main route of the dissemination of events and their evolution over time. According to the statute of the Journalists' Union of Athens Daily Newspapers (ESIEA 1979), the journalists' job objective is the collection, configuration, and presentation of the material published in newspapers, as well on the radio and television stations (Article 5, par. 4, p. 6). As stated in the statute of the Greek Sports Journalists Association (PSAT 2013), sports editors are "those whose main and exclusive profession is the collection, configuration and presentation of sports material, published in the country's daily sports, political and financial newspapers, in sports and of general interest magazines, or presented in the electronic media, the webpages and the news agencies" (Article 6, par. 6). Therefore, journalists working in media organizations specializing in sports reporting actively contribute to social development as producers and distributors of knowledge, mediating the sports reality (Mijatov and Radenović 2019).

Media are the key sources of information and news transmission; in exercising this role, their influence on the prevailing perceptions has been scientifically documented (Spiliopoulos 2020). Therefore, the field of media is empowered to affect people and situations through the topics presented and analyzed, to project specific aspects and eventually to shape the public consciousness (McCombs and Shaw 1972; Vernikou 2019). Sports media, as with any other kind of media within the communication field, hold an important

institutional role in shaping contemporary society's public opinion and dominant concepts (Villamar and Smith 2019). Professional and amateur sports media coverage, of both male and female athletes, has reached unprecedented levels and is growing exponentially (Lewis and Weaver 2015). Therefore, due to and via the ongoing coverage, media and journalists play a key role in the way an issue is shaped (D'Angelo 2019; McCombs 2005).

### 1.1. Successes of Greek National Teams and Media Coverage

The European Basketball Championship first appeared in 1935, and has been held biannually thereafter. The final ranking of the national teams constitutes the criterion for their participation in the final phase of the World Championship (Mundobasket) or the Olympic Games. The Greek national team competed for the first time in 1949, and in 1987 and 2005 won the championship against the Soviet Union and Germany in the final game, respectively (HBF 2021). As anticipated, these successes (as well as the win of the 2004 European football championship) were highly reported not only in the sports but in the political press as well (Vernikou and Mastrogiannakis 2020).

In Greece, a country in which media are a component of most people's daily lives, sports news is emerging as one of the most popular journalistic fields and even comprises a segment of the political press. It is remarkable that the country holds the world record for publishing daily sports newspapers (Spiliopoulos 2020) with more than a thousand specialized sports journalists. Specifically, basketball reporters present a unique characteristic; on the occasions that their coverage, regarding either the clubs or the national basketball team, according to their personal judgment, is perceived as "attacks" to the sport, they unite to protect it in a degree that is considered remarkable (Bourlakis 2019).

In February 2022, five daily sports newspapers were published nationwide; at the same time, a daily and a weekly published newspaper with headquarters in Thessaloniki (the second most populated city of the country) were available in northern Greece (Frontpages 2022), while another one was distributed free of charge in digital form (Sportime 2021). In addition, due to the advent and penetration of the internet, many sports websites have emerged and have been established. Sports news, either in the form of inserts or with a significant number of pages, is also included in almost all the national distributed political newspapers and likewise in the provincial ones.

The following paper is a content analysis that uses framing to examine attribution of responsibility in Greek political and sports publications' coverage of 20 Greek national men's basketball team defeats. This research is important since, to the authors' knowledge, the subject is scarcely investigated globally, and it may constitute a stimulation to other researchers in the field, contribute to the extension of the framing theory, and cover the void in the existing literature.

### 1.2. Framing Theory

A frame placed around a painting is carefully chosen by the artists because it potentially affects the way people view and interpret its substance (Tewksbury and Scheufele 2009). Thus, the visual artist hopes that the public will see the contents of the painting through their own preferable way. Framing in Media works just like a painting frame; by isolating and delimitating the image, it makes it special in relation to others (Kotsanti and Tsigilis 2020). The media's scope is to provide to the public a specific interpretation of a topic; consequently, journalists select the approach of the news presentation to influence the interpretation and evaluation processes of the issues brought to their audience's attention (Ramadan and Prastya 2019).

As Reese (2007) states, the metaphorical concept of framing is credited, by the sociologist Erwing Goffman, to the anthropologist–psychologist Gregory Bateson. Goffman, in his book "Frame Analysis" (Goffman [1974] 1986), dealt with the "organization of (human) experience" (p. 13) and transferred this concept to the social sciences, where it was used for communication analysis (Kreuter 2021). Since then, various scholars have evolved

this concept, considering it as a separate approach to Media research (D'Angelo 2019; D'Angelo and Kuypers 2010; Entman 1993; Gitlin 1980, 2003; Tuchman 1978).

Goffman ([1974] 1986, pp. 21–39) identified two types of frames: (a) the primary frameworks or otherwise natural frameworks, and (b) the secondary or social frameworks (Goffman [1974] 1986, pp. 21–22). In the first case, the primary contexts attempt to administer an essence to events and actions that otherwise would not have any meaning; through these contexts, people interpret what is happening in the world (Davie 2014). In the second case, the secondary contexts are based on the primary, though they present an operational difference (Davie 2014). The sociological foundations of the framing theory were established by Goffman (Tewksbury and Scheufele 2009), and it is one of the most fundamental theories in the communication research field (Kotsanti and Tsigilis 2020; Weaver 2007), and for that reason selected for the present study. However, the two concepts (frame and framing) are not necessarily synonymous (Johnson-Cartee 2005), and therefore a relative clarification is needed for the research objectives. This study adopts Davie's (2014) definition of frame, which refers to the manner in which a topic is presented to the public, while framing is refers to the process by which "a news organization defines and constructs an . . . issue" (Nelson et al. 1997, p. 567).

### 1.3. The Primary Frames in the Media and the Persuasion Processes

According to Tewksbury and Scheufele (2009), the primary frames in communication research present a twofold significance; on the one hand, they are socially constructed categorization systems, used to process information between journalists and between citizens as well, while on the other hand, they are used by the media to influence the targeted audience interpretations. In other words, framing initially shapes and sequentially alters the audience's interpretations and preferences. This is achieved as the frames, in the first phase, are employed to bring the audience's attention to certain events, making them more noticeable, important, or memorable, thus defining the daily agenda (first function/function of the first level of framing). In the second phase, they increase the apparent significance of certain ideas or features of these events (next function/function of the next level of framing). Such an increase in emphasis enhances the likelihood of processing and storing in the memory the meaning of events by the recipients (Entman 1993). In this way, the media activates schemes that encourage the targeted audience to think, feel, and decide in a specific way (Entman 2007; Weaver et al. 2004). Media producers know in advance that this endeavor will be achieved, as people are not well informed and rely on the media to acquire information and knowledge, to cultivate attitudes and create behaviors (Entman 1993). McGuire (1986) has extensively explained the process by which attitudes (that precede behavior) guide our thoughts, choices, and decisions for action. Formation or alteration of attitude is the most frequently studied result of framing. On this basis, Nelson and Oxley (1999) found in their research that the frames in financial news reporting influenced the perceptions of students. In short, the effect of framing is mainly an effect of interpretation. The messages that comprise Media framing are specially designed to change the perceptions or attitudes on specific issues that the Media undertake (D'Angelo 2017).

### 1.4. Sports Journalism and the Effects of Framing

According to Neuman et al. (1992), news frames are "conceptual tools on which Media and individuals rely on to convey, interpret and evaluate information" (p. 60). Therefore, many researchers agree that framing is a necessary tool in news reporting that is used (a) to reduce the content complexity in the aspect of news transmission (Corcoran 2006; Hearns-Branaman 2020; Karlsson and Clerwall 2018; Knight 1999; Marland 2012) and (b) to lead the audience in specific ways of thinking and dealing with situations (Cassino 2007). Sports journalists, as Media "gatekeepers", decide the features and the approach by which athletes and coaches (male and female) are represented in their stories. Thus, news on sports topics and events contains information and frames (Tewksbury and

Scheufele 2009). As a result, this could affect the consumers' thoughts, feelings, and attitudes (in the sense of the degree of favor or disfavor as defined by Eagly and Chaiken (2007) and behaviors (in the sense of responses). Consequently, it could also affect the way consumers will react and/or make subsequent evaluations of sports protagonists (Price et al. 1997). The journalists' judgements regarding the coach and his/her choices of athletes, the performance of the athletes, the decisions of the referees, and also the actions of a team's management members, such as the attitude towards authorities concerning the appointment of referees, the selection of a coach, and the acquisition of players, are presented daily in sports news. Spectators and fans are emotional by nature, and their exposure to media posts, depending on the type of coverage, may affect their future support for the athletes (Lewis and Weaver 2015). Thus, lately, the sports reporters' approaches and processes as they create their stories and the framing they set have been investigated.

Lecheler and De Vreese (2019) named a framing effect as the process by which "a frame in communication affects an individual's frame in thought" (p. 13). A framing effect occurs "when a phrase, image, or statement suggests a particular meaning or interpretation of an issue" (Simon and Jerit 2007, p. 20). However, almost always, the framing effects of the Media are dependent on the basic, individual assessments of the frames (Bazerman 1984; Nelson et al. 1997; Shen and Edwards 2005; van Gorp 2007). Nonetheless, the sports media audience is rarely aware of the presence of frames and the influence they can exert on the creation of their own frames of interpretation (Tewksbury and Scheufele 2009).

### 1.5. The Attribution of Responsibility Frame

Gitlin (2003), while describing media frames, characterized them as "silent, to a great extent" (p. 7) as they are not easily recognizable. He emphasized that they emerge as persistent patterns that combine the selection of topics, their presentation, and the preferred interpretation by the audience that is dependent on the Media news provision. With the use of frames, the journalists initially locate the news, then classify it (Tewksbury and Scheufele 2009), process it with editorial room practices, and make it ready for retransmission to their audience. Sports journalists extensively use such frames that emphasize specific aspects of their stories (Lewis and Weaver 2015); they are the paths through which they choose to highlight and disseminate specific information, in order originally to provoke and afterwards to maximize the audience interest. Due to time and space constraints, as news gatekeepers, they select certain stories for publishing, rejecting others (Shoemaker and Vos 2009). In addition, as they seek to be objective and do not necessarily possess a common understanding of the essence of framing, they often allow communicators or skilled media operators/manipulators to impose (their own) dominant frames in their texts (Entman 1993). Considering that frames "draw the attention to some aspects of reality while obscuring other elements which might lead audiences to have different reactions" (Entman 1993, p. 55), framing theory is applicable to the content of news media (Hearns-Branaman 2020). The present research is based on the theory of framing (Villamar and Smith 2019).

Following the evolution of theory in the field, the authors acknowledged and utilized the newest two-stage model of framing that was processed and proposed by Scheufele (2004). The model proposes that in the first stage, Media framing affects the awareness of the audience, while in the second, it influences the way information is processed (Davie 2014). Therefore, it presents an impact on the judgments, attitudes, opinions, emotions, and decisions of the audience, as recipient of the journalistic messages. Furthermore, the present study adopted, as predetermined frame for investigation, the primary frame of attribution of responsibility of the five-point typology formulated by Semetko and Valkenburg (2000).

This context has been found, by certain studies, to be dominant in the field of sport. Specifically, it has been found in the coverage of sport for development and peace (SDP) by English-language newspapers (Harrison and Boehmer 2020), in the coverage of the

doping phenomenon both in Germany (Starke and Flemming 2017) as well as in Greece (Kotsanti and Tsigilis 2020), and in an organizational crisis at the 2010 Delhi Commonwealth Games (Carey and Mason 2016). In particular, in a similar study to the present one, Dumitriu (2013) studied the existence of the "attribution of responsibility" frame in the 2010 European Women's Handball Championship (7–19 December 2010) and in the 2011 World Women's Handball Championship (2–18 December 2011), examining two general newspapers—Adevarul and Evenimentul Zilei—and two sports newspapers (Gazeta Sporturilor and Prosport1). She concluded that this frame comprises the main strategy of the print media for creating an evaluative position and also for deciding who should be "blamed for the competitive effects" (p. 79).

Therefore, in this research, the existence of "attribution of responsibilities" was investigated, as a primary frame, in the coverage of the defeats of the Greek national men's basketball team. Sports journalists tend to personalize their criticism and, following team defeats, link responsibilities with coaches, players, management, or refereeing. They also pose ethical dilemmas for the acceptance of responsibilities, dismissals or resignations of coaches and executives, as well as the punishment of referees. All the above constitute structural features of the news in sports reporting (Matthes 2009). The specific model and the typology were adopted in the context of a pluralistic approach (D'Angelo and Shaw 2018; Reese 2007), which attributes increased validity and reliability to the study. Moreover, it was adopted since the approaches are compatible with each other, while they provide consistent information for the study of frames (Kreuter 2021). For the scope of the study, frames are comprehended as subtractions that are used to organize or structure the meaning of the message (Davie 2014).

## 1.6. Aim of the Study and Research Questions

Through an extensive literature review (Reese 2007), it was revealed that the relationship between successes in sports and journalism has been researched (Lewis and Weaver 2015; Wenner 2003); however, the case of defeats has not been studied. On this ground, Lewis and Weaver (2015) suggest that the negative sports news should be examined, as it consists of "another area ripe for research" (p. 234). The purpose of this study was to investigate how the Greek media presented the defeats of the Greek men's basketball team in the Eurobasket championships by examining the decade 2007–2017. As suggested by Dumitriu (2013, p. 63), "international sports competitions constitute a resourceful context for analyzing the media framing of responsibility".

In particular, researchers examined the type of framing employed by the Greek print media, starting from the day following each game of the Greek national men's basketball team and continuing up to three days later. As suggested by Dimitrova et al. (2005), despite the large number of studies "there are still gaps in what we know about framing" (p. 25). Thus, the first research question concerned the use of a specific framing by the print media (Semetko and Valkenburg 2000):

RQ1: Does the primary frame of attribution of responsibility appear in the titles and the content of publications?

The second research question focused on the existence of differences in framing among the political publications and also between them and the sports newspapers. Therefore, the second research question was expressed as follows:

RQ2: Are there any differences in the specific framing (attribution of responsibility) among the political publications and also between them and the sports newspapers?

The third research question examined with whom and what the media linked the causes of the defeats:

RQ3: Regarding the defeats, on which causes does the content of the articles focus and with which factors and persons does the Media link the responsibility?

## 2. Materials and Methods

### 2.1. Method Selection

Content analysis was selected as the appropriate research method (Coombs et al. 2017; Entman 1993), which is frequently applied for research in journalism (D'Angelo 2019) since it can be applied to different aspects and types of messages (Weaver 2007). As a research method it is considered extremely flexible and has been widely used for a variety of research purposes and objectives, both quantitative and qualitative, and most, if not all, of the studies on framing and journalism focus on content analysis of media content (Hearns-Branaman 2020). White and Marsh (2006) have defined content analysis as "a systematic, rigorous approach to analyzing documents obtained or generated in the course of research" (p. 22). The articles from the print Media examined by the present research fall under this category. The specific analysis combines reliable findings in the quantitative part with the qualitative method, which is necessary for a deeper understanding of their meaning and interpretation, and this seems to be the ideal approach (Macnamara 2005; Villamar and Smith 2019). The constructivist (ontological and epistemological) assumption of the researchers reinforces and justifies the qualitative part of the method (D'Angelo 2019).

### 2.2. Selection of Approach—Research Validity

Regardless of the approach employed, the reliability and validity of every study is largely dependent on the transparency in the export of the frames, so that they may be considered as "Media frames" instead of "researcher's frames" (Matthes and Kohring 2008, p. 260). Therefore, to avoid being found in front of a methodological "black box" (Matthes and Kohring 2008, p. 262), the present study employed a pluralistic approach, namely, a combination/triangulation of methods (Bryman 2017; Semetko and Valkenburg 2000; Willig 2015).

The deductive approach (Semetko and Valkenburg 2000), which can easily detect differences in framing between different media types, was employed in this study. In our case, the analysis unit was the publication, and the frames were known in advance and were derived from the literature. Specifically, content analysis examined whether in the articles under investigation, the predetermined primary frame of "attribution of responsibility" (Matthes and Kohring 2008, pp. 260–63), which has been repeatedly examined/checked in the literature, was presented.

The deductive approach is applied over time and is unreservedly suggested by many scholars (Camaj 2010; Dimitrova et al. 2005; Igartua et al. 2005; Rendon et al. 2019); Cacciatore et al. (2016) argued that after 40 years of research on framing, researchers are expected to "at least partially operate abductively and explore frames that previous research has indicated as applicable" (p. 14). In this case, it was applied to the field of sport journalism. The researchers, who are specialized both in media and in sports, had a clear idea of the types of frames that might be found in media texts (Semetko and Valkenburg 2000).

To satisfy the study objective, the frame was defined as the method by which the sports sections of the political newspapers, as well as the editorial office of the sports newspaper, investigate, decide, organize, and present the ideas, facts, and topics they cover (Davie 2014). The attribution of responsibility frame introduces an issue with a certain approach so as to attribute responsibility for the cause or its solution either to the State or to an individual or group (Semetko and Valkenburg 2000). Sports journalists, after defeats, link the responsibility with the coach, or with the players, or with the refereeing, and less often with other noncompetitive factors (Lewis and Weaver 2015; SDNA 2021; Newsbeast 2021). Attribution of responsibility is defined as a frame that emphasizes the assumption or attribution of responsibility for a defeat by/to an individual, or more persons.

The researchers also considered the advice of Matthes (2009) that the general frames cannot convey useful information and the term frame must be used carefully. In addition, they appraised/adopted the aspect/directive of Matthes and Kohring (2008), who pointed

out that it is critical that the frames, which are known in advance, be in line with the phenomenon under investigation.

Thus, with a clear/specific picture in mind of the usual types of frames found in media sports texts (Semetko and Valkenburg 2000), the use of a frame that was "tested/checked" in several previous surveys, that of "attribution of responsibility", was selected. This decision, as well as the careful description of the research process, is an essential element to convince the readers of the reliability and validity of the research (Downs 2002). Therefore, the audit trail of the study towards the findings and their extraction is considered necessary and will be reported in detail (Carcary 2009).

### 2.2.1. Material Selection and Recording Procedures

To accomplish the study's objectives, the material was collected from the newspaper archive of the library of the Municipality of Thessaloniki, Greece. The selection criteria of the newspapers were as follows: (a) to be a political or sports newspaper, (b) to be indexed in the municipal library of Thessaloniki with the availability of all the issues for the period 2007–2010, and (c) to have a wide circulation and readability. Based on the above criteria and in combination with the selection of newspapers for the global survey International Sports Press Survey (ISPS) 2011 (Panagiotopoulou 2013), the political newspapers "KATHIMERINI", "TA NEA", and "ETHNOS", as well as the sports newspaper «METROSPORT» were selected. In these four newspapers, 178 reports in total that referred to the defeats of the Greek men's basketball team, up to three days after each game, were identified. The corresponding newspapers were located in the archives, and the articles referring to the 20 defeats of the Greek national basketball team were retrieved. The texts were categorized according to each newspaper, but also by the chronological order by which they were published. The creation of quantitative analysis tables followed, and the procedure was completed with the qualitative analysis of the texts, thus applying a triangulation of methods to increase the validity of the research (Bryman 2017; Willig 2015).

### 2.2.2. Frame Measurements

To measure the apparent extent of the "attribution of responsibility" frame in the newspaper reports succeeding the national team defeats, a scale of 4 items was developed as follows: (a) was a player attributed the responsibility or was he blamed by the journalists? (b) was a coach attributed the responsibility or was he blamed by the media? (c) was an administrative agent attributed the responsibility or was he blamed? and (d) was a referee/secretariat/commissioner/other factor of the match (e.g., international federation, organizing authority) attributed the responsibility or was he blamed? At this point, it must be noted that there were no female coaches during that decade and there were no women journalists that signed the publications under study. In Greece, sports journalism is a stronghold of men (Spiliopoulos et al. 2020), and women's work is limited to covering unpopular sports. The encoder answered "yes" (1) or "no" (0). For a frame to be considered, there needed to be a positive answer (yes (1)) to at least one (or more) of the questions. A higher score (i.e., more positive responses) in the distribution of the scale of attribution of responsibility indicated a higher level of responsibility of the one who is accountable for causing the defeat (Pan and Kosicki 1993). In addition to the above, the selection of that particular frame was made because "in essence, frame analysis examines the selection and salience of certain aspects of an issue by exploring images, stereotypes, metaphors, actors, and messages" (Matthes 2009, p. 349) while, in addition, it aims to provide "more systematic (and) fine-grained knowledge" (p. 359).

### 2.2.3. Reliability Coding

To ensure coding reliability, two coders, who checked 20 randomly selected articles, five from each newspaper, were used. According to Lombard et al. (2002), intercoder reliability, or intercoder agreement, "is a measure of the extent to which independent judges make the same coding decisions in evaluating the characteristics of messages"

(p. 587). Cohen's *k* (Cohen 1960, pp. 39–40) was used to test the reliability/agreement, and the formula for the calculation is k = (Po − Pc)/(N − Pc), where N is the total number of judgments made by each encoder, Po is the observed agreement percentage randomly expected by the encoders, while Pc is the agreement percentage after discussion. The coders agreed for 17 texts, and 3 more decisions were made after discussion, resulting in k = (17 − 3)/(20 − 3) = 0.8235. According to Lombard et al. (2002, p. 600) and Coombs et al. (2017), values above 0.80 suggest high levels of agreement and are acceptable in most situations. Banerjee et al. (1999, p. 6) reinforce the above, stating that values greater than 0.75 can be considered to represent an excellent agreement.

## 3. Results

Prior to the results analysis, it is considered mandatory to clarify the language of the Greek sports texts. Therefore, it has to mentioned that the language itself, and especially the writing on sports, in this case of basketball, is filled with special terms and extensive borrowing of words mainly from the English language, which essentially constitute a "special dialect" (Androulakis 1997, p. 337; 2008). Thus, the journalistic discourse, and consequently the writing of sports news, due to the external interventions it has undergone over time, ends up being polyphonic and loaded with elements of intertextuality (Politis 2001, 2008). Hence, the titles of the print media are discerned for (war) metaphors, cliché phrases, elliptical sentences, grammatical and rhetorical forms, as well as metonyms, in order to state certain situations and actions (Xanthiotis et al. 2020), easily ascertained in the results analysis of the present research. This is why in many cases the title does not seem to be relevant to the text. This comprises common practice of the editorial office and is also a serious disadvantage of sports journalism. The titles are not usually suggested by the text authors; they are set by the editor-in-chief or the director of the Media, who has his own opinion on the facts.

### 3.1. The Attribution of Responsibility Frame

The first research question concerned the detection of the attribution of responsibility frame, which was found in all the print media under study, not only in the content but also in the titles of the articles. It was met 63 times out of the total 178 publications (titles and content) that were investigated, reaching 35.4% (Figure 1).

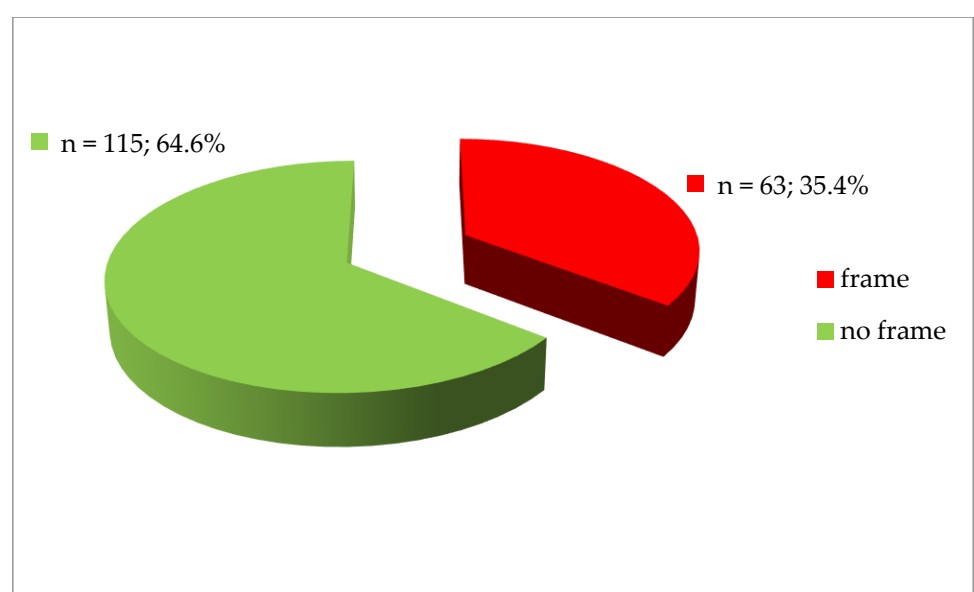

**Figure 1.** Appearance of the "attribution of responsibility" frame in titles/content (overall).

However, the day succeeding the game, the "attribution of responsibility" frame appeared in 45 of the 63 titles and content of articles or other reports, which accounts for 71.4% (Figure 2).

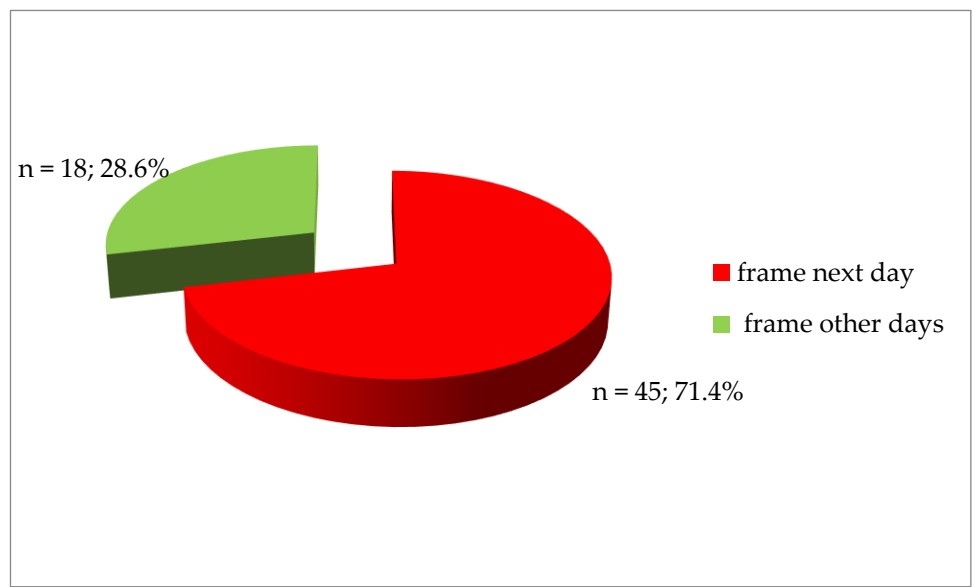

**Figure 2.** Appearance of the "attribution of responsibility" frame in the titles/content the day succeeding the game.

### 3.2. The Differences in Framing between the Newspapers

The second research question of the study concerned the detection of the differences in the articles' framing, between the political newspapers, as well as between them and the sports newspaper. Regarding the differences identified in framing, which were derived from the quantitative analysis, the attribution of responsibility frame was found mainly in the titles and the content of "METROSPORT" articles, both overall (n = 40, 63.5%) and on the day following the game (n = 33, 73.3%). In the political newspapers, it was met to a greater extent in the newspaper "TA NEA", both overall (n = 11, 17.5%) and the following day (n = 9, 20%), and to a smaller degree in "KATHIMERINI", overall (n = 4, 6.3%) as well as the day succeeding the game (n = 1, 2.25%). The newspaper ETHNOS was between the two political newspapers, with the above frame being found in the titles and content of its articles a total of eight times (2.7%), while on the day succeeding the game it was found twice (4.45%), as shown in Table 1.

**Table 1.** Appearance of the "attribution of responsibility" frame in the titles.

| Newspaper | Frame Appearance Frequency (Overall) | Frame Appearance Frequency (the Day after the Game) |
|---|---|---|
| METROSPORT | 40 63.5% | 33 73.3% |
| KATHIMERINI | 4 6.3% | 1 2.25% |
| ETHNOS | 8 12.7% | 2 4.45% |
| TA NEA | 11 17.5% | 9 20% |
| TOTAL | 63 100% | 45 100% |

Overall, the "attribution of responsibility" frame was found in more than six out of ten publications in the sports newspaper METROSPORT (63.5%); that is, ten times more than

the appeared frequency in KATHIMERINI (6.3%), five times more than the corresponding frequency in ETHNOS (12.7%), and four times over the percentage found in the newspaper TA NEA (17.5%). Furthermore, on the succeeding day of the game, it was detected in almost three quarters of the overall number of the articles in METROSPORT (73.3%), which accounts for triple times over the total number of all the other political newspapers articles. Specifically, the ratio of the publications where the "attribution of responsibility" frame was found in METROSPORT compared to KATHIMERINI was 33/1, compared to ETHNOS was 33/2, and compared to TA NEA was 33/9. Finally, an important difference that was found between the political publications is that in the newspaper TA NEA, the overall appearance of the frame under investigation was the same as the additive appearance in the other two political newspapers (11/12), while for the succeeding day of the match, the frame was found three times more often than in the other two newspapers additively (9/3).

### 3.3. Defeat Causes Focused on, and Agents and Individuals Blamed by the Media

The third research question examined the defeat causes that the content of the articles were focused on and the agents and individuals linked to relevant responsibilities by the Media. It has to be noted that, for the purposes of the present study, the titles of the newspapers have been translated by the authors, from the Greek language to the English language. In this regard, Figgou (2020) pointed out that "unavoidably, a translation involves the danger of losing subtleties of meaning" (p. 206). The qualitative analysis of the publications front-pages revealed that the causes and the responsibilities were linked with (the classification is made according to the significance order as traced in the reports):

(1) The players' weaknesses in the game (METROSPORT 2007a).
(2) The players' mistakes in the game (METROSPORT 2009b, 2017a).
(3) The fatigue and the loss of strength by the athletes (KATHIMERINI 2017; METROSPORT 2009c; TA NEA 2007a).
(4) The players' stress (METROSPORT 2013b).
(5) The viruses that afflicted the players (ETHNOS 2007).
(6) The losses of players caused by injuries (METROSPORT 2011).
(7) The athletes' decision not to participate in the championship (KATHIMERINI 2011).
(8) The lack of:

    a. Focus of the basketball players in the game (METROSPORT 2013c, 2017c);
    b. Faith in their abilities by the athletes (METROSPORT 2015);
    c. Passion in their game (METROSPORT 2017b).

(9) The coaches' mistakes (ETHNOS 2013; METROSPORT 2007b).
(10) Bad refereeing (METROSPORT 2009a).
(11) Off-field factors, such as:

    a. The uproar in the stadium (METROSPORT 2007c);
    b. The bad scheduling of the championship that deprived the players of rest (TA NEA 2007b);
    c. Even in . . . superstitions! (METROSPORT 2007d; TA NEA 2015).

(12) The wrong handlings by the Greek Basketball Federation administration members, such as:

    a. The coach selection (METROSPORT 2013a);
    b. Other decisions and actions both by its president George Vasilakopoulos[1] and by the general secretary Panagiotis Tsagronis[2], such as the improper scheduling/bad scheduling (TA NEA 2017).

### 3.4. Other Frames and the Framing Function

Although the purpose of the research was to study the existence of a specific framework, that of "attribution of responsibility", during the analysis the researchers constantly reflected that by applying the predefined frames, there was no assurance that other important frameworks that could emerge from an inductive analysis would not be omitted

([Matthes and Kohring 2008](#); [Semetko and Valkenburg 2000](#)). Thus, the existence of other frames was identified; they are presented here, in an effort to reduce the weaknesses of the research. An important finding was that in the articles that were published before the games, as a pre-announcement of them, no hypertonicity of Greek nationality was found, as it happened after the victories ([ETHNOS 2005](#)); even after the defeats, the reports praised Greekness. Furthermore, the coverage of the defeats suffered by the Greek national team, by the national teams of neighboring countries, such as FYROM[3], were not constrained by METROSPORT to the sporting event, but also referred to the national and political disputes of Greece with each country (e.g., Turkey and FYROM).

The other frames that were identified were the following: audience involvement (see uproar), an athlete's individual action (e.g., the praise of Antetokounmpo, Papaloukas[4]), goals and ambitions (such as qualifying for the national team), consequences (i.e., exclusion from the continuation of the games), human interest (e.g., the way the national team competed in order to choose an opponent for the next stage of the tournament ([Vetakis 2009](#), p. 46), conflict (i.e., various disputes of the team), attention deflection and distraction (e.g., emphasis on wrong-doings or stealing of the ball), and health (focusing on athletes' health effects individually and over the team). Finally, through the coding of the attribution of responsibility frame, all four basic functions of the framework proposed by [Entman](#) ([1993](#), p. 52; [2007](#)) emerged, in relation to their nature ([Camaj 2010](#)).

These functions are as follows: (1) the problem definition, such as defeat/victory, (2) the diagnosing of causes that explain the reasons for qualifying or exclusion from the continuation of the games after a defeat/victory, (3) the moral judgments when there is criticism of the athletes' and coaches' performance, (4) the treatment recommendation by sports editors and media in order to deal with a situation, such as the removal of the coach or the attention that should be paid by the Greek Basketball Association in the forthcoming events.

## 4. Discussion

The purpose of this research was to investigate the framing of the Greek national men's basketball team defeats in the European championships in the decade 2007–2017, by the Greek Press. Towards this end, the content of 178 texts was located and analyzed in the light of the framing theory, in relation to (a) the primary frame of "attribution of responsibility", (b) the differences among the political newspapers as well as between the political and sports newspapers, and (c) the causes that the articles focused on and the agents and individuals where/with whom they linked the responsibilities. Moreover, as it emerged during the analysis in the journalistic texts, other frameworks were met as well, while the framing functions according to [Entman](#) ([1993](#), [2007](#)) were also found.

The first research question investigated whether the "attribution of responsibility" frame was observed/appeared in the journalistic articles. This frame was observed in all publications, especially the day succeeding a defeat. It is anticipated that, the succeeding day of a game, the journalists who covered it wrote their text immediately after its end, while they had still fresh memories and impressions, to criticize and evaluate the defeat in the media. The text of the reporting was then submitted for publication, in combination with the statements made by the coach and the players after the end of the match, both in the press conference and in the mixed zone. Frequently, players and coaches, in their account of the defeat, either took responsibility for the various situations that led them to the loss of the match or took their own responsibility for the negative end result, which journalists considered when writing texts/reports.

According to unpublished data of the International Sports Press Survey (ISPS) 2021, conducted in Greece, the coverage of a match published the next day covers a little less than the one-third of the total content of the sports pages in a political newspaper (28.12%) and is the primary "main factor" of the content of the articles. This explains the high presence of the "attribution of responsibility" frame, which was observed in all media texts, mainly the next day of the match.

As the days go by from the end of a game and as the next game approaches, the journalistic interest in a game decreases significantly. Therefore, the assessment of the defeat is not the dominant news element for the media. For this reason, the "attribution of responsibility" frame was found to be significantly reduced on the second and third day after the defeat (18.6%).

The articles published on a game's eve, and on the day of its conduct, include the preparation of the team for the (next) game, any absences of players, and especially its announcement. According to ISPS 2021 for Greece, the pre-announcement of matches covers 19.7% of the total content of the sports pages of a political newspaper and is the third "main factor" of their articles content. The importance of the game's pre-announcement is shown by another finding of the same study, as it constitutes the secondary "main factor" the succeeding day of a game, covering 14.75% of the newspaper material. Therefore, as the day of the game approaches, more space is devoted by the press to its pre-announcement, and therefore the appearance of the "attribution of responsibility" frame on the second and third day after the defeat is limited.

The "attribution of responsibility" frame emerged as a recurring pattern in journalistic texts, combining the audience's preferred interpretation, which is based on the media for news acquisition, as described by Gitlin (2003). The finding is consistent with Tewksbury and Scheufele's (2009) proposals that news on sports and events contains information and frames and reinforces the findings of Starke and Flemming (2017) and Kotsanti and Tsigilis (2020) on doping coverage in Germany and Greece, respectively. It is also supported by the findings of Dumitriu (2013) in a similar sports environment, the European Women's Handball Championship. It seems that the "attribution of responsibility" frame is, for Greece as well, the main strategy of the print media, which aims to influence the processes by which the public interprets and evaluates the issues raised (Ramadan and Prastya 2019) so as to consequently decide who is to blame for the negative results of the national basketball team (Dumitriu 2013).

In addition, the validity of the finding is enhanced by the satisfactory intercoder reliability found in Cohen's $k = 0.8235$ (Semetko and Valkenburg 2000), as well as by the careful and in-depth description of the "attribution of responsibility" frame (Downs 2002).

### 4.1. Differences in Coverage

The investigation of the existence of differences in the framing, between the political and sports newspapers as well as among the political newspapers, concerned the second research question. The differences identified are typical.

The attribution of responsibility frame was found mainly in the titles and in the content of "METROSPORT" sports newspaper articles, both overall and on the day following the game as well. In the political newspapers, it was met to a greater extent in the newspaper "TA NEA", both overall and the following day, and to a smaller degree in "KATHIMERINI", overall as well as the day succeeding the game. The results on the newspaper ETHNOS were placed between the other two political newspapers, with the above-mentioned frame being found in the titles and content of its articles eight times overall, while on the day succeeding the game, it was found two times.

Overall, the "attribution of responsibility" frame was found in more than six out of ten publications in the sports magazine METROSPORT (63.5%); that is, ten times more than the appeared frequency in KATHIMERINI, five times more than the corresponding frequency in ETHNOS, and four times over the percentage found in the newspaper TA NEA. Furthermore, on the day succeeding the match, it was detected in almost three quarters of the overall number of the articles in METROSPORT (73.3%), which accounts for triple times the total number of all the other political newspapers articles. Specifically, the ratio of the publications where the "attribution of responsibility" frame was found in METROSPORT compared to KATHIMERINI was 33/1, compared to ETHNOS was 33/2, and compared to TA NEA was 33/9. Initially, this can be explained by the kind of the publication, as it is a purely sports newspaper. As such, a sport newspaper can dedicate

multiple space both on its front page and on its internal pages, to highlight the issues of the Greek national basketball team. Usually, the result of the national team game is the central theme on the front page and the dominant theme on more than one internal page, with the quotations of several independent texts (articles). For example, in addition to the coverage of the game, there are special columns that deal exclusively with basketball as well as articles/commentary columns that escort the games. In these cases, captions over and under the articles, as well as those on the photos, along with the journalists' opinion, many of whom claim to be fans of the national team, may often include the context under investigation. On the contrary, political newspapers rarely devote space on the front page to host the result of a sport event, especially when it is unfavorable. Equally limited space is occupied by the game reporting on their internal pages. Consequently, the appearance of the "attribution of responsibility" frame is less often.

An important variation that was found between the political publications is that in the newspaper TA NEA, the frame under investigation was met in total as many times as in the other two newspapers additively (11/12), while in the next day of the game, the frame was met triple times compared to its appearance in the other two newspapers additively (9/3). This can be explained by the fact that the newspaper TA NEA had a special sport insert named "OMADA (TEAM)", with several pages dedicated to basketball and the national team games. Of course, the newspaper ETHNOS also had a special sport insert (ETHNOSPORT) in which the frame was found to a satisfactory degree (n = 8, 12.7%); however, not on the next day following the game (n = 2). This constitutes an opposing finding of the investigation and can be explained by the fact that the newspaper content was finalized early, the journalists had already delivered their articles and there was no time to analyze the match; thus, the analysis apparently was performed on the second or third day after the conduct of the game. This finding is consistent with Shoemaker and Vos (2009) argument that the journalists, as news gatekeepers, due to time and space constraints, select a certain number of stories for transmission, while rejecting others.

The common finding was that the newspapers always blamed someone for the loss of the victory. The "attribution of responsibility" frame in all four publications was basically heading in the same direction. However, in contrast to the political ones, the sports newspaper METROSPORT deepened the competing part and referred to the coach's Trinkeri selection of only one system to be applied by the team during the 2013 games. Furthermore, the accusations of the sports medium against the Greek Basketball Association administration for its selections before the championship were intense. On the contrary, only one political newspaper referred to the Hellenic Basketball Federation.

The findings are supported by Dumitriu (2013), who suggested that the coverage of the negative issues in the media "seems to be more rational, highlighting a more critical and argumentative reason". Furthermore, they are in line with the sports editors' principles of ethics for the practice of journalism "conscientiously and in good faith" (Statute of the Greek Sports Journalists Association, article 8, par. A and par. B), complying strictly with its basic principles, which are the independence and freedom of the press, to ensure the full information is provided to the citizens (Statute of the Journalists' Union of Athens Daily Newspapers, article 2).

### 4.2. The Focus of the Articles on the Defeat Causes and the Agents and Individuals the Responsibilities Were Linked with

The Media frame also invites people to use the information and concepts offered by the journalists to interpret the issues/topics they are concerned with (Tewksbury and Scheufele 2009). This explains the fact that to the larger extent, the attribution of responsibility to agents and individuals investigated by the third research question referred to the mistakes of coaches, players, and referees. To a lesser extent, responsibilities were linked with the players' illnesses and absences, and with the improper scheduling of the host country as well as with the erroneous actions of the Greek Basketball Association administration prior to the event.

This diversity arises because illnesses and absences were not encountered in every defeat; however, the responsibilities for the loss of the victory were linked with players, coaches, and referees every time after the defeat. All these accusations have one thing in common: they were made by the journalists on the day immediately after the end of the games, and the reports they prepared were published in the newspapers the succeeding day. On the contrary, the accusations against the host country for bad scheduling, which led to the exhaustion of the Greek athletes due to fatigue, appeared in reports at the end of the championships, and these complaints were made by the athletes themselves. Finally, the statement/evaluation of the administration's actions was performed according to the judgment of the journalists, also a posteriori and as a consequence of the negative results. With regard to the framing of the Media in relation to the "attribution of responsibility" frame, Dumitriu (2013) has suggested "a personalization effect, which is addressed mainly to individual players" (p. 79).

As the responsibility for provoking or solving social problems could be attributed to the individual (athlete or coach), to members of the Greek Basketball Federation administration, or to the organizing authority, when studying the texts it was observed that a considerable part of them link the responsibility for the defeats with specific individuals following confrontational journalism (Johnson-Cartee 2005). Many of the published articles were not limited to the transmission of facts and commentary, but took a clear stance, in favor of one side, or against the other. There were articles that blamed the defeats on the coaches, and on specific athletes too, attributing various derogatory descriptions to them, or asking for their removal from the national team.

For example, the basketball player Papaloukas was characterized as "prone to mistakes", and his teammate Diamantidis as "hesitant", while the team's coach, Trinchieri[5], was blamed for his decision to choose only one system that the team applied during the games throughout the tournament, and the issue of dismissal was directly raised (ETHNOS 2013). In the case of athletes, the findings are consistent with the research of Lewis and Weaver (2015), who noticed that the performance of athletes in the games interests the sports audience and is an important element of journalists' stories, while in the case of coach Trinikeri, they are in line with the findings of Dumitriu (2013) that the media with the practice of personalization builds up "scapegoats" when negative results occur.

In some articles, the attribution of responsibilities concerned the refereeing of the matches, which was accused of favoring the opposing teams. The host country was also blamed for bad scheduling of the games, which led to lack of rest for the Greek athletes, meaning the national team did not have the necessary breaks needed to relax, unlike its opponents. As suggested by D'Angelo (2019), frames, through continuous media coverage, offer to the audience interpretive schemes for the comprehension of the articles. This seems to have been performed by the Greek journalists who covered the games for their newspapers.

Furthermore, in 2007, an article that blamed the defeat on the uproar that existed on the court was recorded. This finding became more understandable in the football World Cup that followed in 2010 in South Africa, when the sound of the famous vuvuzela disturbed the athletes (Bairaktaris 2014). Later, this event led the Federation International of Basketball Associations (FIBA) to ban vouvouzelas from the Mundobasket Championship held in Turkey in 2018 (STAR 2018).

In addition, the alleviation of responsibilities for the national team was via the illnesses faced by the players, and also the absences of some of so-called "key members", which forced the Greek national team to play in the championship with significant shortcomings (METROSPORT 2007c). The attribution of responsibilities also included the Greek Basketball Federation administration and especially the president and the general secretary, who were accused of making wrong choices regarding the selection of coach Trinkeri, and in general of bad planning in view of the games. This finding is also supported by Dumitriu (2013), who identified a corresponding imputation of responsibility to the Ro-

manian Handball Federation, for the defeat of the women's national team in the 2011 European Championship.

The two-step model of framing (Scheufele 2004), in which it is stated that frames constitute methods that the sports editors choose in order to highlight and disseminate specific information on issues, initially by provoking and then by maximizing the interest of the audience, interprets the case of the athletes, of the coach, and of the executives of the Association administration as well. Although the study did not investigate the impact on the sports audience, it cannot be ignored that Media framing in the first phase affects the awareness of the public, while in the second it affects the way information is processed (Davie 2014). Thus, it has an impact on the judgments, attitudes, opinions, feelings, and decisions of the audience, which is the recipient of the journalistic messages.

Especially in the case of athletes, since it has been documented that the frames aim to change attitudes and consequently behavior (Nelson and Oxley 1999), athletes' statements and the attribution of responsibilities by them to noncompetitive factors could affect the thoughts and feelings of the audience, and hence the degree of sympathy towards them (Eagly and Chaiken 2007). The fact that journalists have published athletes' statements linking responsibility with noncompetitive factors could potentially affect the way that the audience will react and/or make subsequent evaluations of the sports protagonists (Price et al. 1997).

In conclusion, the focus of the articles that linked the responsibility for defeat to a large number of actors and individuals "explains the team's failure, as cumulus of all these negative inputs" (Dumitriu 2013, p. 79).

*4.3. Other Frames and Framing Functions*

The hypertonicity of Greek nationality after the victories of the National team, the reference of METROSPORT to the national disputes of Greece with respective countries (Turkey and FYROM), relates to the primary frame of culture/nationalism, which was identified in the research of Alabaster (2017), which was conducted in Japan. In the sports field, and particularly in the sport of basketball, the specific frame was found as a primary frame in the research of Ličen and Billings (2013), who examined the framing of the Slovenian journalists' televised speech for their national team, in the organization of Eurobasket 2007. A similar trend was observed in a study regarding basketball games as well, by Ličen and Topič (2008), for the televised speech on the Euroleague organization at the level of professional clubs, uttered by both the Slovenian and the Croatian journalists.

The frames, which were also found to exist in sports journalism in Greece and which have been identified by other researchers in journalistic texts (Coombs et al. 2017; Dimitrova et al. 2005; Mikelonis 2017; Robeers 2019), fall under the following categories: (a) "public participation" (effect of noise from the crowd on the stadium stands), (b) "individual action" related to the performance of athletes in the game, (c) "goals and ambitions" for the continuation of the games or the future of the national team, (d) "consequences" from the exclusion of the remaining games, (e) "human interest", such as the team's deliberate performance in selecting an opponent in the next game, (f) "conflict", such as the various disputes that arose in the team after conflicting statements by athletes (TA NEA 2017, p. 15), (g) "deviation and distraction" from the players during the match that leads to mistakes, and (h) "health", such as illnesses and injuries of athletes with effects on the team's performance. The four framing functions (Entman 2007) have also been found in previous sports media studies, such as those of Kozman (2017), Lewis and Weaver (2015), and Smith and Pegoraro (2020).

According to the literature review, the use of frames is intentional, as media content producers offer to the public preferred interpretations (Entman 1993) of the news, as opposed to the audience who is unaware of their existence (Tewksbury and Scheufele 2009). However, most of the Greeks sports journalists are not university graduates of Journalism Departments so they lack theoretical/academic knowledge of the frame functions (Matsiola et al. 2022; Spiliopoulos 2007). The professional experience of the first researcher

led to the conclusion that they "feel"/"sense" them more empirically, rather than being aware of or perceiving them, which contradicts the literature. The Greek journalists do not intentionally use frames/framing, they do it empirically, or rather, instinctively, as this is a practice of the editorial room and is taught by the older journalist to the younger.

As a future study subject, it would be important to compare the coverage approaches by the Press to the successes and failures of the Greek national men's basketball team, and also to expand the research to other popular team sports, such as football. In addition, the research should include the morphological characteristics (extent) and the practice (tone) of the coverage. Moreover, by examining articles in digital media, which in recent years tend to constitute the majority, the research could lead to safer conclusions regarding the subject of the study, in the above or any other predefined frameworks, as encountered in the present study. It is also proposed to investigate more analytically the operational mechanisms of framing, as well as the inductive coding of articles, to determine the kind of frames that may arise (Bell and Hartman 2018; Semetko and Valkenburg 2000). Finally, research on the impact of framing on the audience, which was outside the scope of the present work, should be included in future research in order to gain the most integrated insight of the subject under investigation.

## 5. Conclusions

In conclusion, "attribution of responsibility" is a common phenomenon in journalism, and a basic (primary) frame in the media, forged in the editorial room (D'Angelo 2019). For this reason, it was chosen to be examined in the present study, which proved that the phenomenon is also observed in sports journalism, in all of the print media under study. This finding agrees with the study on doping by Kotsanti and Tsigilis (2020) in Greece, but also with other international research, as mentioned in the theoretical section of the study (Carey and Mason 2016; Harrison and Boehmer 2020; Semetko and Valkenburg 2000), and is an important finding for the field of sports journalism in general. Sports journalists are often accused of being/seen as "cheerleaders" of the team and not as serious as other journalists; even the kind of news they deal with is characterized as soft compared to the hard news of political reporting (English 2017).

The articles that were studied showed that the journalists chose to frame the defeats by attributing responsibilities primarily to the players, the coaches, the Federation administration, and the host country. On a second level, the responsibility was linked with the players' illnesses and injuries ("health" frame), and with external, in relation to the competition, factors such as the unbearable noise on the field from the vuvuzela ("public participation" frame). The ascertainment of the attribution of responsibility frame in all the studied media suggests, on the one hand, the importance and potential impact of sports culture on the framing of news issues (Semetko and Valkenburg 2000), while on the other hand, the presence of the other frames ("individual action", "goals and ambitions", "consequences", "human interest", "conflict"), as well as the framing functions (Entman 1993, 2007), is suggested, in the international literature, to guide the audience in preferable interpretations (Lewis and Weaver 2015).

Nevertheless, the general evaluation of the publications related to basketball in Greece should be made with caution, since the reporters of the specific sport are characterized as "Masons" in the slang of Greek sports editors. This is appointed to them in the sense that "with a conscious choice in the interest of basketball" they act as members of a "stoa" (Apodytiriakias 2016), hiding many of the wrongdoings. In the harsh occasions or in the cases that according to their personal judgment are perceived as "attacks" in the sport, they unite, creating ad hoc alliances in order to protect it (Bourlakis 2019). This alludes to the "Bedouin Syndrome" (Dunning et al. 1986, p. 230)[6], and as a phenomenon or as a pattern of behavior, it is particularly encountered in the Greek basketball reporters; therefore, they are perceived by the sports audience as acrobats between (hall of) fame (https://doubleteam.gr/hall-of-fame/ accessed on 20 February 2022) and (hall of) shame (https://doubleteam.gr/hall-of-shame/ accessed on 20 February 2022). This uniqueness

regarding Greek basketball journalists is in accordance with the stereotypes of other sports journalists (English 2017), but perhaps in a worse way.

The present study employed the predefined "attribution of responsibility" frame to investigate whether it existed in the texts examined. This would involve a relative limitation, as Matthes and Kohring (2008) have identified and warned about, if the research did not identify other important frames or did not identify the four basic framing functions as posed by Entman (1993, 2007) and as met in other studies (Kozman 2017; Lewis and Weaver 2015; Smith and Pegoraro 2020). The researchers had a clear idea of the probable types of frames present in the articles, and although they chose to investigate the primary frame of the "attribution of responsibility", at the end they did not overlook other equally or less important frames (Semetko and Valkenburg 2000), thus drastically decreasing this limitation.

**Author Contributions:** Conceptualization, N.T. and P.S.; methodology, P.S., N.T. and I.T.; software, P.S., M.M. and I.T.; validation, N.T. and P.S.; formal analysis, P.S., N.T. and I.T.; investigation, I.T.; data curation, P.S., I.T., N.T. and M.M.; writing—original draft preparation, P.S., M.M. and N.T.; writing—review and editing, P.S., M.M. and N.T.; visualization, P.S. and M.M.; supervision, N.T. and P.S.; project administration, N.T. All authors have read and agreed to the published version of the manuscript.

**Funding:** This research received no external funding.

**Institutional Review Board Statement:** Not applicable.

**Informed Consent Statement:** Not applicable.

**Data Availability Statement:** Data are available upon request to the authors.

**Conflicts of Interest:** The authors declare no conflict of interest.

## Notes

[1]    George Vasilakopoulos was the president of the Greek Basketball Federation.
[2]    Panagiotis Tsagronis was the General Secretary of the Greek Basketball Federation.
[3]    This is the country with the current name Republic of North Macedonia, as the Former Yugoslav Republic of Macedonia (FYROM) was renamed with the Prespa agreement on 12 June 2018, a name which it used in the event studied in this paper.
[4]    Giannis Antetokounmpo and Theodoros Papaloukas were basketball players of the Greek National Team in 2009 Eurobasket Championship
[5]    Andrea Trinchieri was the National Team coach in the 2013 Eurobasket Championship.
[6]    As "Bedouin Syndrome" in football, Paul Harrison (1974) refers to the tendency to form ad hoc alliances according to the principles: a friend's friend is a friend, an enemy's enemy is a friend, an enemy's friend is an enemy, a friend's enemy is an enemy.

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
