# Peer review of "The Framing of the National Men’s Basketball Team Defeats in the Eurobasket Championships (2007–2017) by the Greek Press"

_journalmedia, doi:10.3390/journalmedia3020023_

Round 1

Reviewer 1 Report

Row 143: spelling mistake -  "members -such"

Row 144: spelling mistake - "players-, are"

Row 309: "(a metropolitan city in northern 309 Greece)" - unnecessary

Row 528: "ISPS 2021" - it is unclear what the abbreviation means

Row 561:  ".8235" - It is unclear what these numbers stand for.

Author Response

Reviewer: 1

Comments and Suggestions for Authors

Row 143: spelling mistake -  "members -such"

Row 144: spelling mistake - "players-, are"

Row 309: "(a metropolitan city in northern 309 Greece)" - unnecessary

Row 528: "ISPS 2021" - it is unclear what the abbreviation means

Row 561:  ".8235" - It is unclear what these numbers stand for.

Answer: The authors would like to thank the reviewer for his/her interest in their manuscript. They have considered his/her comments and suggestions and in the revised manuscript they were all addressed.

Rows 143 and 144: the dashes were removed

Row 309: the explanation in the bracket was removed

Row 528: the ISPS abbreviation has been presented in row 314 in full, however for the readers’ convenience we further added in Row 528 as well.

Row 561: the .8235 is the value of the Cohen's k which was used to test the intercoder reliability. In the revised manuscript it is mentioned again by its name in the respective row. 

Reviewer 2 Report

The paper is a content analysis that, through the theoretical lens of framing, examines attribution of responsibility in coverage of 20 Greek national men’s basketball team losses, up to three days after these loses. The study also explores differences between the political and sports publications, and who or what journalists credited as being responsible for the loses. The author(s) found that the attribution of responsibility frame was in all publications, more heavily in Metrosport articles than the political publications (i.e., Kathimerini, Ta Nea, and Ethnos). The study qualitatively showed 12 factors and 8 sub-factors attributed to the defeat, including players’ weaknesses, mistakes, fatigue, and stress.

The author(s) give a thorough overview of framing research and backs up its arguments with a wide range of relevant studies.

The paper’s premise is important because it shows when and on what sports journalists are most likely to place responsibility for loses. This is of particular interest in Greece and beyond because sports journalists are often seen as “cheerleaders” for the team and not as serious as other journalists. The author(s) noted a unique characteristic of Greek basketball journalists that I did not fully understand. I could not determine if this group does not fit the stereotypes of other sports journalists, or if it does, but perhaps in worse ways? This uniqueness about sports journalists, whether it’s Greek journalists or otherwise, should be played more prominently in the introduction. This really is a special element of the study.

The paper’s method is explained well, as is the theoretical importance of content analyses explained. I don’t usually see people explain qualitative and quantitative content analyses so well.

There were some areas that confused me. I was not sure if this confusion was a theoretical problem, or a language structure problem. For example, I was confused by the definitions of frames and framing. The definitions used in the methods section differed from the literature review. How were the four items were developed? Were they seen in past studies? Also, say more about the difference between the primary and secondary contexts (p. 2). Were both of these examined in the paper?

On page 3, section 1.3 The primary frames in the media and the persuasion processes: Do journalists “use” frames “to influence the target audience interpretations” intentionally? How conscious of this do you believe Greek journalists are of frames? When you refer to “the media,” do you mean journalists/people creating content? For example, on page 3, line 114: “the media activate schemes that encourage the targeted audience to think, feel and decide in a specific way.” Is this a media effects claim? Or should “the media” be “journalists” or “content creators”?

A framing/sample question: “He” or “his” is predominantly used when referring to coaches (p. 3, line 141; p. 7, line 330), players (p. 7, line 329), administrative agents p. 7, line 131), editor-in-chief or the director of the media (p. 8, line 374), for example. Should I assume there are no female coaches, players, administrative agents, directors or editor-in-chiefs that would have been cited for referred to in the study?

For the research questions, I wasn’t sure how the first and third research questions differ. How was the attribution of responsibility frame “found in all under study publications” but “was met 63 times out of the total 178 publications”? Wouldn’t it have been found in all 178 publications? And for the third research question: Are these listed in any particular order? For example, was “the players’ weaknesses in the game” most prevalent out of all the themes? And is the example listed after the theme exhaustive? (For example, in 1.) The players’ weaknesses in the game (“One speed back” METROSPORT 6.9.2007) – was this date and story the only place this theme arose, or is this just one example?

State in the introduction what the paper is going to do. Give the reader a “road map” of what is about to happen and why it is important. For example, the last paragraph could begin with, “The following paper is a content analysis that uses framing to examine attribution of responsibility in Greek political and sports publications’ coverage of 20 Greek national men’s basketball team losses. …”

Overall, this paper has promise and compelling results. It was enjoyable to read and pulls together a wealth of past research on which it has built its arguments. Some of the confusion may be sentence structure issues. An editor could help relieve this and make the author(s) points clearer.

Author Response

Reviewer: 2

The paper is a content analysis that, through the theoretical lens of framing, examines attribution of responsibility in coverage of 20 Greek national men’s basketball team losses, up to three days after these loses. The study also explores differences between the political and sports publications, and who or what journalists credited as being responsible for the loses. The author(s) found that the attribution of responsibility frame was in all publications, more heavily in Metrosport articles than the political publications (i.e., Kathimerini, Ta Nea, and Ethnos). The study qualitatively showed 12 factors and 8 sub-factors attributed to the defeat, including players’ weaknesses, mistakes, fatigue, and stress.

The author(s) give a thorough overview of framing research and backs up its arguments with a wide range of relevant studies.

Answer: We are thankful for the positive feedback of the reviewer along with the respective constructive comments, which were taken into consideration in the revised manuscript.

The paper’s premise is important because it shows when and on what sports journalists are most likely to place responsibility for loses. This is of particular interest in Greece and beyond because sports journalists are often seen as “cheerleaders” for the team and not as serious as other journalists.

Answer: The authors would like to thank the reviewer for further pointing out this issue, his/her remark was included in the conclusion section of the revised manuscript.

(Pages 16-17). This is an important finding for the field of sports journalism in general, as sports journalists are often accused of being / seen as "cheerleaders" of the team and not as serious as other journalists; even the kind of news they deal with is characterized as soft compared to the hard news of political reporting (English, 2017).

The author(s) noted a unique characteristic of Greek basketball journalists that I did not fully understand. I could not determine if this group does not fit the stereotypes of other sports journalists, or if it does, but perhaps in worse ways? This uniqueness about sports journalists, whether it’s Greek journalists or otherwise, should be played more prominently in the introduction. This really is a special element of the study.

Answer: The reviewer’s comment was elaborated further in the introduction and the conclusions section of the revised manuscript.

(Page 2). It is remarkable that the country holds the world record for publishing daily sports newspapers with more than a thousand specialized sports journalists (Spiliopoulos, 2020). Specifically, the basketball reporters present a unique characteristic; in the harsh occasions which according to their personal judgment are perceived as "attacks" to the sport, either regarding clubs’ coverage or on the Basketball National Team coverage, they unite to protect it in a degree that is considered remarkable (Bourlakis, 2019).

(Page 17). This uniqueness regarding Greek basketball journalists is in accordance with the stereotypes of other sports journalists, perhaps in a worse way.

The paper’s method is explained well, as is the theoretical importance of content analyses explained. I don’t usually see people explain qualitative and quantitative content analyses so well.

Answer: The authors would like to thank the reviewer for his/her appreciation in their work.

There were some areas that confused me. I was not sure if this confusion was a theoretical problem, or a language structure problem. For example, I was confused by the definitions of frames and framing. The definitions used in the methods section differed from the literature review.

Answer: According to the authors’ judgment, in the literature review, the difference between frame and framing was clarified and various definitions from different authors, maybe even from scientific fields, were quoted.

The definitions in the methodology section are the ones that the authors chose to employ for the needs of the present work. If they seem to be different, it is probably because of the adaptation, perhaps even the mixing of definitions that the authors made during their scientific judgment to fit the field of journalism, for the needs of research.

How were the four items were developed? Were they seen in past studies?

Answer: The four frames, meaning the four functions of the frames according to Entman (if we clearly understand the reviewer’s comment), are secondary findings that were not among the original aims of the research. However, they were added to the results to reduce the probable weaknesses of the research. They were not further explained, as well as the other frames which were identified; only the basic frame of attribution of responsibility was deployed.

The four functions have obviously been found in previous studies such as Entman 2007, Kozman, 2017, Lewis and Weaver, 2015, Smith and Pegoraro, 2020.

Also, say more about the difference between the primary and secondary contexts (p. 2). Were both of these examined in the paper?

Answer: The secondary frames were not investigated, as it was beyond the scope of the work. The reference to the existence of primary and secondary frames was made to show that we are aware of this distinction and the role of each level (first or second) and to clarify that this research would be on the primary context of "attribution of responsibility". After all, this is a first attempt to investigate the framing of the Greek press in sports texts. In future studies, in other sports or media, we intend to examine these frames as well. Therefore, in our judgment and precisely for these reasons, we consider that no further deepening is needed unless the reviewer deems it necessary.

On page 3, section 1.3 The primary frames in the media and the persuasion processes: Do journalists “use” frames “to influence the target audience interpretations” intentionally? How conscious of this do you believe Greek journalists are of frames?

Answer:  According to the literature review the use of frames is intentional, as media content producers offer to the public preferred interpretations (Entman, 1993) of the news, as opposed to the audience who is unaware of their existence (Tewksbury and Scheufele, 2009). However, most of the Greeks sports journalists are not university graduates of Journalism Departments so they lack theoretical / academic knowledge of the frame functions (Spiliopoulos, 2007; Matsiola et al. 2022). The professional experience of the first researcher leads to the conclusion that they "feel" / "sense" them more empirically, rather than they are aware of or perceive them, a fact that contradicts with the literature. The Greek journalists do not intentionally use frames / framing, they do it empirically, or rather, instinctively as this is a practice of the editorial room and is taught by the older journalist to the younger. This text has been added in the revised manuscript as well in the discussion section since the authors estimate that it should follow the analysis performed (page 16-17).

However, this idea of the reviewer is a very good one to study further on the subject! The authors would really appreciate if there were relevant references on the subject that the reviewer would like to share!

When you refer to “the media,” do you mean journalists/people creating content? For example, on page 3, line 114: “the media activate schemes that encourage the targeted audience to think, feel and decide in a specific way.” Is this a media effects claim? Or should “the media” be “journalists” or “content creators”?

Answer: In the studied literature, this is not specified. Our study regarding the Media was performed in the sense that the Media is made up of people who produce content, and this is the result of the Media.

A framing/sample question: “He” or “his” is predominantly used when referring to coaches (p. 3, line 141; p. 7, line 330), players (p. 7, line 329), administrative agents p. 7, line 131), editor-in-chief or the director of the media (p. 8, line 374), for example. Should I assume there are no female coaches, players, administrative agents, directors or editor-in-chiefs that would have been cited for referred to in the study?

Answer: The reviewer’s assumption is right! In the corresponding parts of the literature review the feminine antonym has been added (page 3). However, there were no female coaches during that decade and there are no women journalists that sign the publications under study. Besides, in Greece the sports journalism is a stronghold of men (Spiliopoulos et al. 2020) and their work is limited to covering unpopular sports. If there were women signing the reports, we would clarify that we use the masculine gender for the needs of the study as we have done in our previous work (Matsiola et al. 2022).

This text has been added in the revised manuscript as well (page 7).

For the research questions, I wasn’t sure how the first and third research questions differ.

Answer: The reviewer’s comment made the authors rethink the phrasing of the RQs. The first research question is general and examines whether the attribution of responsibility frame is present. The third one is more specific and investigates the persons / factors / causes the attribution of responsibility is associated with. The third research question was corrected as follows:

(Page 5). Regarding the defeats, on which causes, does the content of the articles focus on and with which factors and persons do the Media link the responsibility?

How was the attribution of responsibility frame “found in all under study publications” but “was met 63 times out of the total 178 publications”? Wouldn’t it have been found in all 178 publications?

Answer: The authors would like to thank the reviewer for pointing out this phrasing mistake which was due to the translation in English. By using the word “publications” we meant “all print media” under study, meaning all the newspapers. This is the reason that the attribution of responsibility frame was not met in all 178 texts/reports.

And for the third research question: Are these listed in any particular order? For example, was “the players’ weaknesses in the game” most prevalent out of all the themes? And is the example listed after the theme exhaustive? (For example, in 1.) The players’ weaknesses in the game (“One speed back” METROSPORT 6.9.2007) – was this date and story the only place this theme arose, or is this just one example?

Answer: The authors would like to thank the reviewer for providing the chance to clarify this part of their research. The classification was made according to the significance order as traced in the texts. Furthermore, the examples are not exhaustive, but illustrative. The topics were composed by a series of reports that were not included due to (probable) lack of space.

State in the introduction what the paper is going to do. Give the reader a “road map” of what is about to happen and why it is important. For example, the last paragraph could begin with, “The following paper is a content analysis that uses framing to examine attribution of responsibility in Greek political and sports publications’ coverage of 20 Greek national men’s basketball team losses. …”

Answer: The authors would like to thank the reviewer for his/her suggestion which was considered very interesting and helpful for the potential readers; thus, they added the following paragraph in the revised manuscript.

(Page 2). The following paper is a content analysis that uses framing to examine attribution of responsibility in Greek political and sports publications’ coverage of 20 Greek national men’s basketball team defeats. This research is important since, to the authors’ knowledge, the subject is scarcely investigated globally, and it may constitute a stimulation to other researchers in the field and it will contribute to the extension of the framing theory and will cover the void in the existing literature.

Overall, this paper has promise and compelling results. It was enjoyable to read and pulls together a wealth of past research on which it has built its arguments. Some of the confusion may be sentence structure issues. An editor could help relieve this and make the author(s) points clearer.
